



# Incorporation of Airborne Wind Energy Systems to Enhance Resiliency for a Microgrid in Rural Puerto Rico

Nico E. Galarza Morales[1], Jimmy E. Quiroz[1], Victor D. Garcia[1], Evan G. Sproul[1],
Rachid Darbali-Zamora[1], and Brent C. Houchens[1]

[1]Sandia National Laboratories, Albuquerque, New Mexico, USA

**Correspondence:** Nico E. Galarza Morales (nico.galarza@sandia.gov) and Brent C. Houchens (brent.houchens@sandia.gov)

**Abstract.** This study evaluates the deployment potential of airborne wind energy systems (AWES) in Puerto Rico to identify early-adopter locations and assess impacts to microgrid resilience. Prospective sites across the island are assessed in quantum global information software using infrastructure and environmental layers combined with high-altitude wind resource data. Culebra, an island reliant on an underwater transmission line and highly susceptible to hurricanes, is selected as a representative

case study for microgrid modeling. A real-world published power curve for a commercial 120-kW AWES, in combination with local wind and solar resource data, are integrated into the Microgrid Design Toolkit to simulate standalone and hybrid systems incorporating AWES, photovoltaics, and battery energy storage systems under realistic outage conditions and design-basis threats such as tropical storms and hurricanes. Seasonal complementarity between wind and solar is assessed, and performance metrics are evaluated with an emphasis on resilience outcomes. Results demonstrate that AWES can support a combination

of priority and non-priority loads during extended grid disruptions and enable faster post-storm re-energization in isolated or infrastructure-limited settings, establishing Puerto Rico as a strong candidate for early-stage AWES adoption. Optimized results show that a configuration of three AWES systems with battery storage achieved approximately 92% and 91% energy availability for non-priority and priority loads respectively during modeled outages, while a hybrid configuration integrating one AWES, a photovoltaic array, and one battery energy storage system yielded approximately 85% and 91% availability for

non-priority and priority loads, respectively.

## 1   Introduction

Airborne wind energy systems (AWES) are an emerging technology in wind power capable of harvesting stronger and more persistent winds at higher altitudes, with the potential to improve energy resilience in environments not viable for traditional towered-wind energy generators (Weber et al., 2021; Faggiani and Schmehl, 2018). The viability of high altitude wind power

has been demonstrated through multiple regional resource assessments and site studies, including detailed technoeconomic and Global Information Software (GIS) based analyses in various locations such as Northern Ireland (Lunney et al., 2017), the Middle East (Yip et al., 2017), and various country-wide site identification efforts for Europe such as Coca-Tagarro (2025). These studies demonstrate that wind resources at operational AWES altitudes showcase higher persistence and lower seasonal variability compared to lower elevation winds.





The integration of AWES in hybrid and off-grid power systems has also been assessed in various studies. Reuchlin et al. (2023) developed and validated a detailed AWES, photovoltaic (PV), battery energy storage system (BESS), and diesel hybrid system sizing model for remote applications, where results demonstrated significant reductions in levelized cost of electricity compared to diesel-only systems. Hybrid systems, integrating AWES and PV into microgrids have further been explored for constrained environments, including off-world applications such as Martian habitats, where AWES were shown to complement

solar power and short-term storage to sustain continuous electrical demand under low-solar irradiance conditions (Schmehl et al., 2024). Together, these studies establish AWES as a deployable generation technology capable of integration into various microgrid configurations and hybrid renewable plants across different settings.

Despite the growing number of AWES resource assessments and hybrid system integration cases, few studies have examined the integration of AWES in microgrids within hurricane-prone regions using real-world grid reliability constraints.

Most existing studies focus on mainland regions, moderate wind climates, or purely economic optimization, with a limited scope on extreme-weather driven outage behavior and utility-scale reliability benchmarks. This gap is particularly relevant for hurricane-prone locations such as the Caribbean islands, particularly island grids such as Puerto Rico.

This work evaluates the performance of AWES based microgrid configurations for a representative rural community in Culebra, Puerto Rico, under various outage scenarios based on observed grid reliability metrics and storm events specific to

the selected location. A wind-only configuration including multiple AWES and a BESS is compared with a hybrid system configuration integrating AWES, PV, and BESS. The abilities of the various configurations to meet priority and non-priority loads during grid outages are assessed. By integrating AWES within a realistic island-grid environment exposed to extreme weather, this study extends previous AWES microgrid research with a resilience focused outcome.

## 1.1   Drivers for Potential Early-Adopter AWES Markets

Current AWES are modest in nameplate capacity, achieving generation in the 100-200 kW range. Furthermore, no farms of AWES have yet been deployed. Thus, communities with modest power needs that could be met by a few AWES are very attractive to advance the technology. Such deployments would facilitate integration testing of multiple AWES to consider open questions including optimum flight paths of multiple systems, cycle phasing between systems, wake effects, and design of hybrid microgrids with storage.

Because AWES have yet to be produced in high volume, the current levelized cost of electricity (LCOE) they generate is still high compared to typical electricity rates in industrialized countries. But long-term projections suggest even small (order 100-kW) fully commercialized AWES could have an LCOE of around 0.12 Euro/kWh ($0.14/kWh) (Faggiani and Schmehl, 2018). Larger MW-scale AWES could theoretically have lower LCOE than even towered wind, though MW-scale AWES are yet unproven. Thus, communities currently paying high rates for electricity, often the case on islands, are attractive as

AWES early-adopters because these systems will offer competitive LCOE in the near-term. Presumably advances in the kite technology will allow relatively low cost re-powering of AWES in the future.

Finally, AWES offer resiliency benefits for challenging environments. Compared to towered wind, AWES can be landed during storms or hurricanes and redeployed quickly after. Currently the ground stations for several 100-kW systems are built





into shipping containers anchored to the ground, providing storm resistant storage. This modularity also offers the potential for
fast-response deployment options to provide power for disaster relief operations.

## 1.2   Case Study for Culebra, Puerto Rico

Puerto Rico, a U.S. territory in the northeast Caribbean Sea, offers a promising prospect for such testing. The 179-km (~111-
mile) long main island and smaller surrounding islands including Vieques and Culebra, the latter circled in Figure 1, is well
known for its longstanding electrical reliability challenges, high energy rates (currently standing 29.6 % above the U.S. national
average), and frequent tropical storm and hurricane exposure.

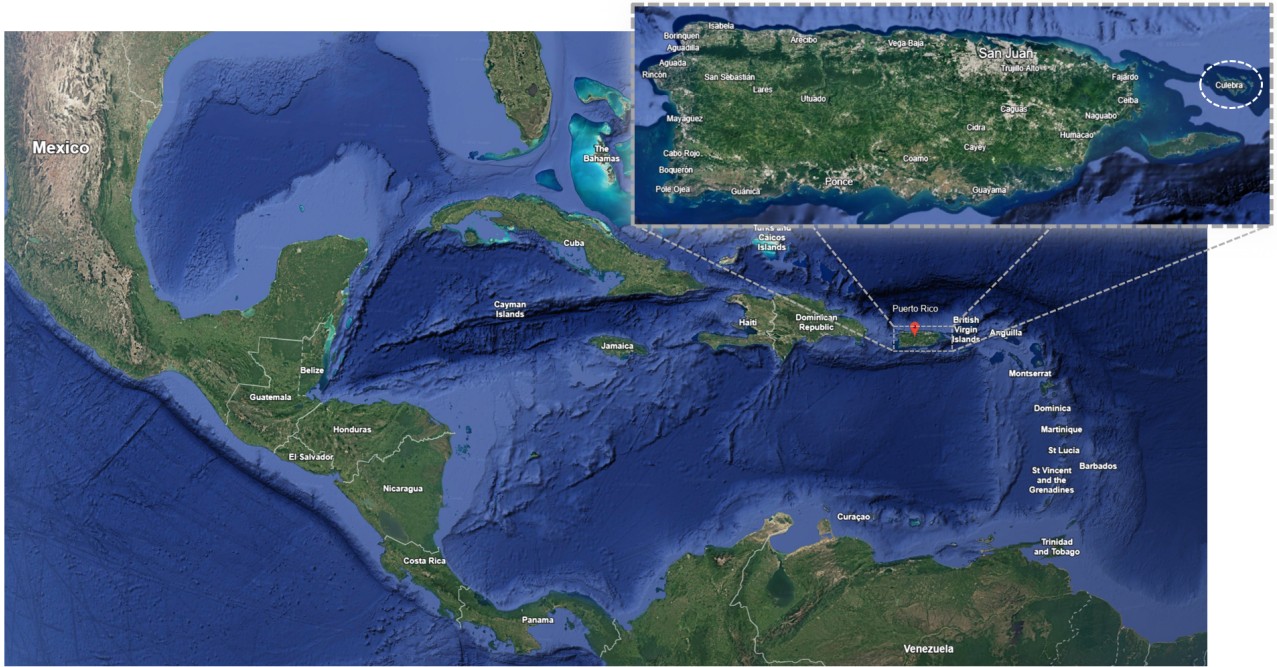

**Figure 1.** Location of Puerto Rico in the Caribbean Sea, with the smaller island Culebra called out. Image © Google Earth 2021.

High hurricane risk has proven challenging for traditional towered wind energy technologies deployed on the island. Such is
the case in the Punta Lima wind farm in Ceiba, Puerto Rico. Commencing operation in the year 2012, this wind farm provided
enough energy to meet the demand for 9,000 homes in different towns near the eastern coast. However, it was destroyed with
the passage of category 5 hurricane Maria in 2017, after just 5 years of service (PREPA, 2019). This wind farm was later rebuilt,
resuming operation in March 2024 (Polaris Renewable Energy Inc., 2024). This example provides a clear representation of the
high risk setting of the islands, offering a realistic stress test environment for newly emerging AWE technologies that feature
rapid deployability and storage features.





This study evaluates the viability of implementing AWES into a microgrid for a rural region in Puerto Rico as a strategy to enhance energy availability and therefore energy resilience in a hurricane prone environment. A real system with a published power curve is used to provide the most realistic assessment of a possible near-term deployment. The cost of electricity could be competitive with current prices and reliability of the system is considered relative to current grid stability challenges. Unlike towered wind turbines, AWES can be landed before extreme weather events and redeployed immediately afterward, offering an advantage for locations with frequent storm or hurricane events. Additionally, some AWES feature a containerized design, eliminating the need for large foundations, facilitating transport and deployment. This assessment considers various microgrid configurations integrating AWES, solar PV, and battery storage, using the Microgrid Design Toolkit (MDT) (Eddy and Gilletly, 2020), implementing real representative load data, a representative AWES power curve, and design-based threats specific to the island such as storms and power outages.

## 2 Puerto Rico Energy Profile, Grid, Hazards and Renewable Resources

This study considers a potential early adopter market of Culebra, a small island community shown in the inset of Figure 1. Though grid tied to Puerto Rico by an undersea transmission line, Culebra has a highly vulnerable power grid. Storms and hurricanes cause outages and the cost of electricity is high. Thus self-sufficiency for critical loads at a competitive price is a high priority for Culebra. In this way, this case study is representative of many small island communities or island nations. To motivate this case study, it is first useful to understand the broader picture of Puerto Rico's electricity rates and grid stability metrics as discussed in Section 2.1. The added hazards that further challenge electricity delivery in Puerto Rico are discussed in Section 2.2. Possible resiliency solutions are dependent on the wind and solar resources described in Sections 2.3 and 2.4.

### 2.1 Energy Rates, Grid Stability and Hazards in Puerto Rico

The average cost of energy in Puerto Rico is high. The national average electricity rate in the USA is approximately 17.47 ¢/kWh; however, in Puerto Rico the average electricity rate for residential consumers is approximately 22.64 ¢/kWh (U.S. Energy Information Administration, 2025b). Therefore, the electricity cost on the island for non-commercial residents is about 30% higher than the national average.

Furthermore, the electric grid has complex challenges with a history of financial and maintenance lag, resulting in a system that often falls behind accepted reliability metrics. Since June 1, 2021, the privately-owned Luma Energy has been responsible for power distribution and transmission across the entirety of Puerto Rico (LUMA Energy, LLC, 2021). According to the FY2023 Resolution and Order on Luma's Metrics Summary Report (Puerto Rico Energy Bureau (PREB), 2023), the Bureau sets annual benchmarks and minimum performance standards for Lumas's reliability indices. These benchmarks serve as performance goals, and are used to assess fulfillment of the operators contractual obligations, outside of major storm events qualifying as force majeure. As shown in Figures 2-4, the reported monthly System Average Interruption Duration Index (SAIDI), System Average Interruption Frequency Index (SAIFI), and Customer Average Interruption Duration Index (CAIDI) are significantly higher than that of the U.S. average levels (U.S. Energy Information Administration, 2025a). The benchmarks





established by PREB for FY2023 were 1,242 minutes for SAIDI, 10.6 interruptions per customer for SAIFI, and 117 minutes for CAIDI. Luma achieved notable reductions in SAIDI and SAIFI, successfully meeting the Bureau's benchmark thresholds for both indices. These improvements represent a significant reduction in both the duration and frequency of service interruptions across the island. However, the CAIDI metric remains well above the benchmark value, indicating that while outages are occurring less frequently, the average restoration times are still longer than the Bureau's target levels.

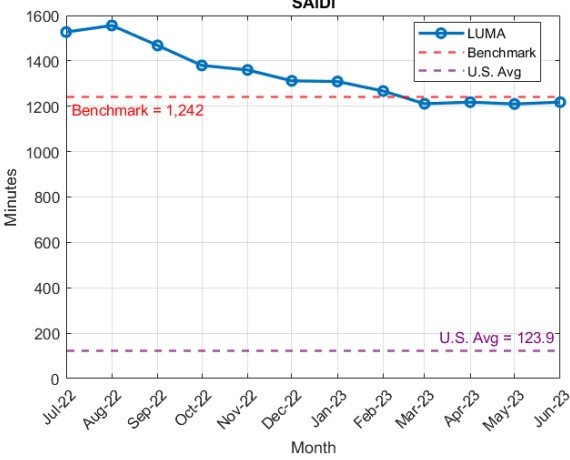

**Figure 2.** Puerto Rico system average interruption duration index FY2023, benchmark values set by PREB FY2023 (Puerto Rico Energy Bureau (PREB), 2023), and U.S. 2023 average (U.S. Energy Information Administration, 2025a).

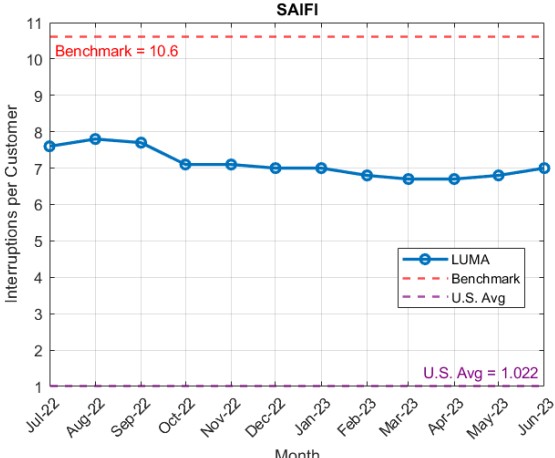

**Figure 3.** Puerto Rico system average interruption frequency index FY2023, benchmark values set by PREB FY2023 (Puerto Rico Energy Bureau (PREB), 2023), and U.S. 2023 average (U.S. Energy Information Administration, 2025a).

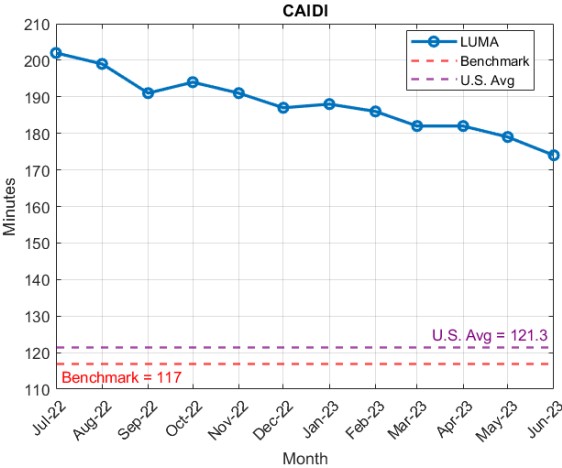

**Figure 4.** Puerto Rico customer average interruption duration index FY2023, benchmark values set by PREB FY2023 (Puerto Rico Energy Bureau (PREB), 2023), and U.S. 2023 average (U.S. Energy Information Administration, 2025a).

While these results express the overall system level performance of Puerto Rico's electric grid without including major events, it is important to note that many remote regions, including Culebra, are disproportionally impacted by additional factors such as severe weather. Hard to access or particularly high risk areas, highlighted in Figure 5, would benefit the most from the integration of distributed energy sources such as AWES.

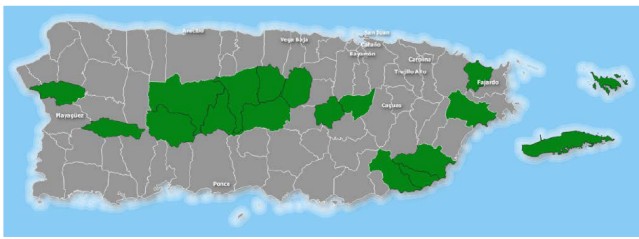

**Figure 5.** Map of identified remote municipalities in Puerto Rico (reproduced from Baggu et al. (2024)).

## 2.2   Hazards for Puerto Rico

Puerto Rico's location in the Caribbean Sea also makes it susceptible to powerful storms resulting in sometimes significant force majeure grid outages not captured in SAIDI, SAIFI and CAIDI metrics. For example, the island experienced one of the longest power outages in the history of the U.S. after Category 5 hurricane Maria in 2017 (Baggu et al., 2024). As shown in Figure 6, Puerto Rico was impacted by three Category 4 or 5 hurricanes and many weaker storms between 2005 and 2020.

      Along with other infrastructure damage, a towered wind farm was destroyed in hurricane Maria and took many years to
bring back online. Thus the need for storm-resilient energy generation is clear, particularly for critical loads such as hospitals and clinics.



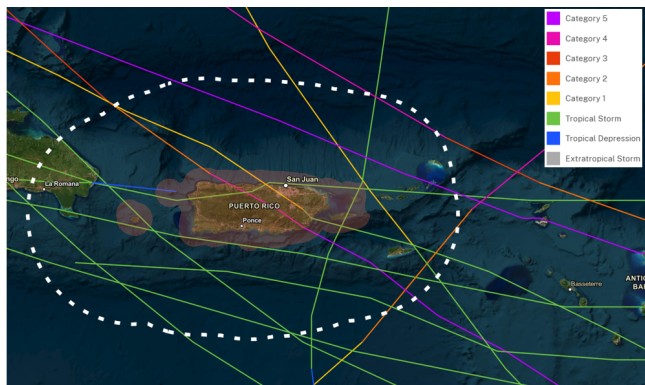

**Figure 6.** 15 year-long Puerto Rico tropical storm/depression and hurricane trajectory history 2005-2020, NOAA.

## 2.3 Wind Resource for Puerto Rico

It is useful to consider the wind resource for Puerto Rico at various altitudes. Figure 7 reproduces results from a National
Renewable Energy Laboratory (NREL) study of the 20-year mean wind resource in Puerto Rico (Sengupta et al., 2022) based on
WRF simulations with models selected to best align with an array of NOAA National Data Buoy Center (NDBC) observations.

The wind resource below 40-m is almost universally poor, making smaller kW-scale towered wind turbines ineffective. At
120-m and above, in the range of large utility-scale towered wind turbines, several regions demonstrate commercially viable
wind resources (Sengupta et al., 2022). This is particularly true in southern coastal regions. Other regions near the center of
the island also demonstrate potentially useful wind speeds at these altitudes, however much of this terrain is mountainous and
difficult to access. Additionally, Vieques and Culebra, smaller islands located off the east coast of the main island, demonstrate
promising wind speeds for altitudes of 120-m and above, but are extremely exposed to storms. The Global Wind Atlas (GWA)
also provides the annual average mean power density available in the wind, the wind direction frequency, and annual average
wind speed for various altitudes based on ERA5 data. The trends of the GWA reported wind speeds at 200-m elevation generally
agree very well with those in Figure 7, though the GWA resolution offshore is less refined compared to the NREL study,
particularly in the wind shadow behind the main island. The significant improvement of the wind resource with altitude suggests
potentially favorable conditions for AWES deployment above 200-m. The containerized nature of currently commercialized
AWES makes them especially attractive for deployment near remote population centers.

## 2.4 Solar Resource for Puerto Rico

The tropical setting of Puerto Rico ensures that it has relatively consistent daylight hours all year, offering a similarly consistent
solar PV resource. The wind resource improves for most of Puerto Rico overnight (Yang et al., 2023). The complementarity of
the hourly solar resource with the wind resource (each averaged over all days of the year) is shown for Culebra in Figure 8. The
wind and solar resource exhibit almost perfect complementarity suggesting a hybrid wind-solar system could likely be viable.
This motivates a comparison of AWES-BESS (wind-only) and AWES-PV-BESS (hybrid) systems.





**Figure 7.** 20-year-mean wind speed contour plots based on WRF modeling at NREL for elevations ranging from 10 to 200-m (reproduced from Sengupta et al. (2022)).





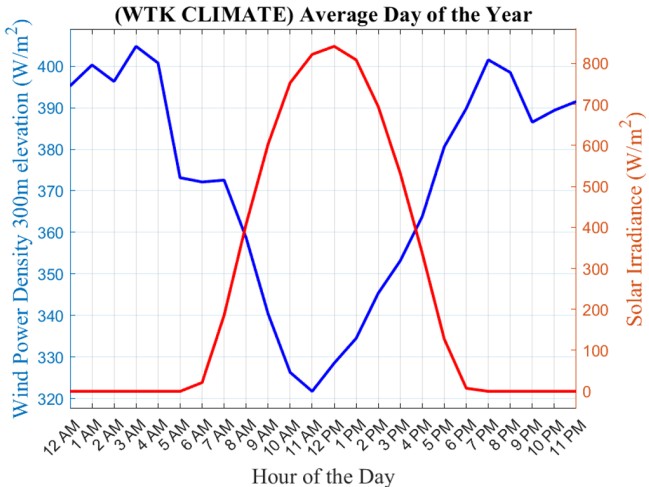

**Figure 8.** Culebra average day of the year wind and solar resources highlighting the hourly complementarity. Note the difference scales.

## 2.5 Selection of Culebra for Case Study

Considering the wind resource and storm frequency, deployment of towered wind is likely to remain challenging in Puerto Rico. However, airborne wind could be attractive given the improved wind resource above 100-m altitude, and the ability to land the system during storms and then redeploy quickly. A high-impact, near-term deployment scenario would have modest power needs matching the current nameplate ratings of commercialized AWES, in a region with a highly vulnerable grid.

    Given these considerations, Culebra appears well suited for AWES deployment. Using GIS layers shown in Figure 9, the

island was assessed to identify critical elements such as protected habitats, military bases, and airports that could hinder or benefit deploying AWES on the island. Though grid tied to the main island of Puerto Rico, Culebra is also on the front line of storms making is susceptible to damage at either end of the subsea transmission cable. A potentially desirable location for the placement of the AWES would be in the vicinity of the hospital (clinic), which is modeled as a critical load. This is also near the landing point of the transmission line from the main island. A deployment west of the hospital would allow the kite or kites

to fly over the water as the dominant wind direction is from the east.

    Figure 10 provides the mean wind speeds at 200-meter altitude for Culebra from GWA, consistent with values from Figure 7. Thus Culebra features a promising wind resource at the AWES operating range with 8.29 m/s average annual wind speeds and an extremely consistent direction. Figure 11 shows the average wind speed for all hours of a day, averaged over the year, for various altitudes typical for AWE flight from the NREL WTK-Climate database.



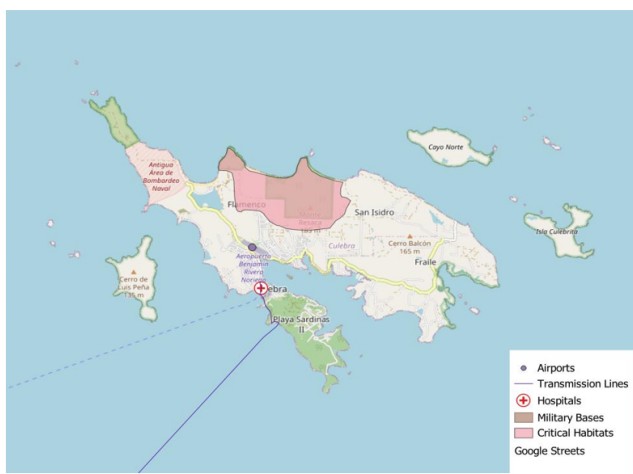

**Figure 9.** Culebra GIS layers identifying elements of interest such as airports, transmission lines, hospitals, and critical habitats. https://www.openstreetmap.org/copyright © OpenStreetMap contributors 2025. Distributed under the Open Data Commons Open Database License (ODbL) v1.0.

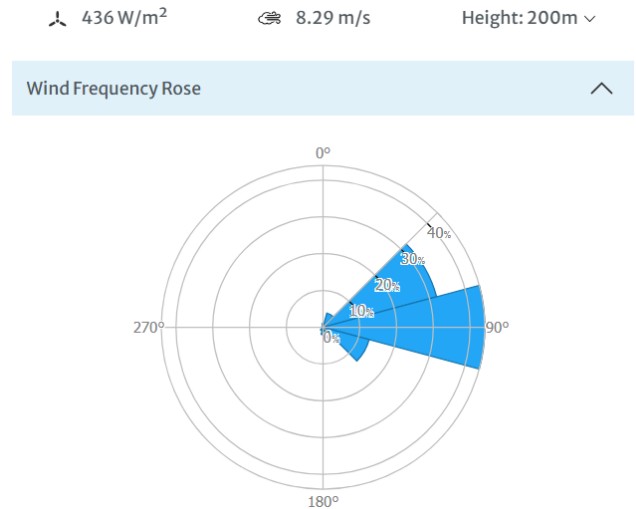

**Figure 10.** Global Wind Atlas average annual wind power density, average annual wind speed, and wind direction frequency (wind rose) at 200-meter elevation for Culebra, 18.315578°, -65.292687°. https://globalwindatlas.info/en/

Using real load data obtained from a small representative community, a portion of Culebra is modeled including 34 residential loads and one larger critical load representing a small health clinic. Figure 12 depicts the trends for residential and critical loads separately showing a residential peak during later hours of the day. The critical load peaks in the early morning and decreases during the overnight hours.



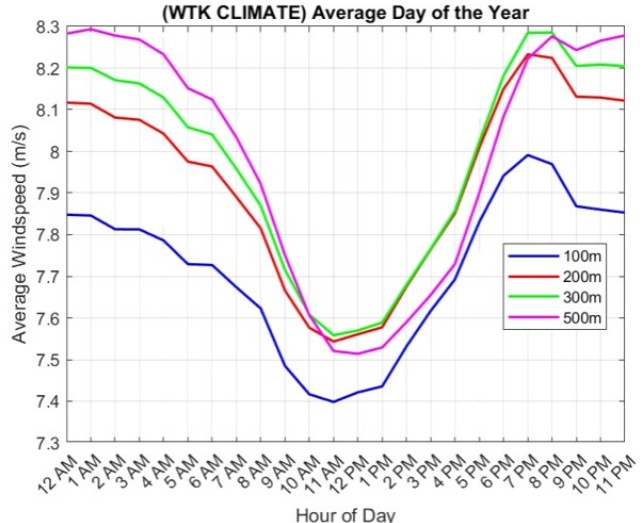

**Figure 11.** WTK-Climate data average day of the year wind speed at various elevations for Culebra.

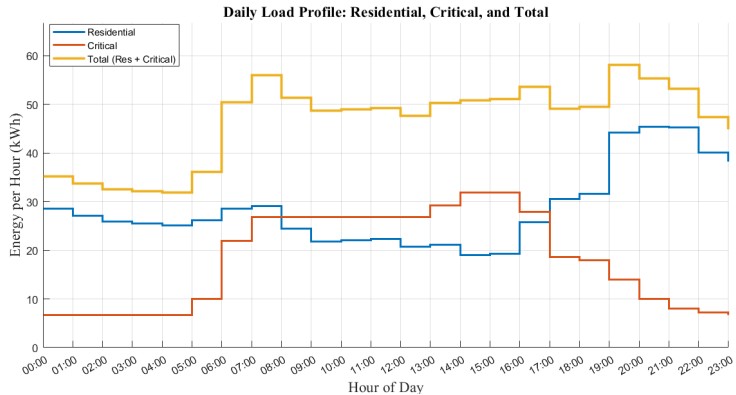

**Figure 12.** Daily average residential, critical, and total load trends for a representative rural community in Puerto Rico.

## 3 AWES and Microgrid Modeling


The existing AWES considered in this study is introduced and characterized in Section 3.1. The power curve is presented and assumed losses are described. The technoeconomic modeling approach is then described considering the cost of a pilot project. Then, MDT modeling considerations are described in Section 3.2. This includes the operating wind envelope, loads and target performance metrics, and storm and other threats.





### 3.1 AWES Specifications and Assumptions

This case study investigated the integration of the SkySails Venyo (PN-14) system into microgrid applications for Culebra. Various optimized systems were identified using MDT software developed by Sandia National Laboratories (Eddy and Gilletly, 2020). To create a system profile that represents an AWES, system-specific parameters such as rated capacity and potential failure modes must be assessed and included in MDT to represent the system performance under site-specific conditions. The Venyo (PN-14), shown in Figure 13 is a containerized high altitude wind energy system developed by SkySails Power with a 120-kW rated capacity capable of operating at up to 600-meter elevations.

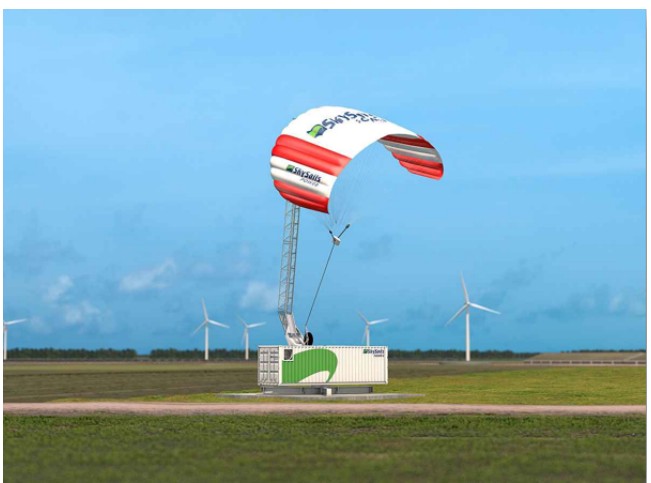

**Figure 13.** Skysails (PN-14) Venyo airborne wind energy system. https://skysails-power.com/

The independently measured, published power curve data shown in Figure 14 was used to fit a representative function for energy output calculations based on hourly wind data. The polynomial fit shown was used in calculations, with a cut-in of 5 m/s and cutout assumed at 20 m/s as a conservative upper limit to avoid damage during high winds. Note this is more conservative than the manufacturer's published limit of 25 m/s, but doesn't affect the results presented for Culebra as all wind events above 20 m/s are also above 25 m/s on an hourly average.





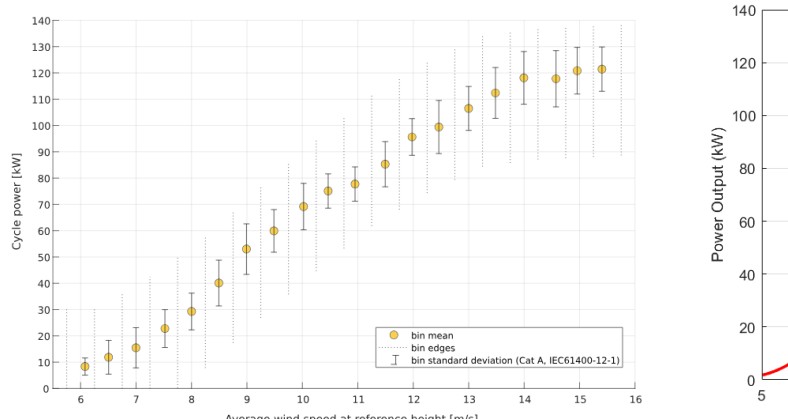
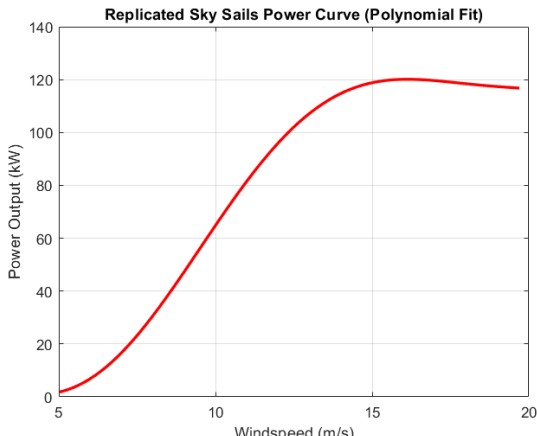

**Figure 14.** Published power curve by Skysails for the (PN-14) Venyo airborne wind energy system (Bartsch et al., 2024), and modeled curve fit used to calculate energy production.

Hourly wind speed data from NREL's WTK-Climate database was investigated at 100, 200, 300 and 500-m altitudes for Culebra (Draxl et al., 2024). It was found that between 300-m and 500-m, the wind speed data varied by less than 3% on average as shown in Figure 11. Thus, although AWES can operate at varying elevations, hourly wind data from a fixed operating altitude of 300-meters was used for the modeling.

### 3.1.1 AWES Operational Uncertainties

System losses are not well understood or characterized for AWES. Because the technology is just reaching commercialization stage, maintenance costs are relatively unknown, but presumably will be relatively high for initial deployments and decrease quickly as OEMs and operators gain experience. Here it is assumed that systems operate unattended with minimal human intervention to decide when to land and redeploy. This concept has been demonstrated, but is not widely tested.

To start to account for these uncertainties, this study assumes a range from an optimistic 18% total system losses (similar to the losses expected for PV or towered wind), to conservative 27% total system losses, 50% larger than the optimistic scenario. Culebra island showed a estimated capacity factor ranging from approximately 18 to 20% operating at 300-meters. It is important to note that system losses might be higher than the proposed assumption, especially due to the nature of energy production via an automated pumping cycle. After performing the cross-flow figure eight energy production cycle while reeling out, the system uses a fraction of the energy produced to reel in the kite and prepare for another energy production cycle. The power curve is dependent on the required frequency of reel-in phases.

The hourly wind speed from WTK-Climate, averaged over 7 years of data, along with the power curve, predicts the power produced for each hour of the year. This is a conservative approach since averaging the hourly winds tends to reduce the duration of generation at rated power. Then the power produced is reduced by applying the optimistic and conservative losses.





The sum over all 8,760 hours of each year provides the annual energy production (AEP) of the AWES. Comparison of the AEP to the available energy in the wind over the year gives the capacity factor.

### 3.1.2 Technoeconomic Considerations

A target LCOE of ¢26/kWh is used to set a cost benchmark for back calculating a CapEx for the system, deployed in Culebra. This target LCOE, provided by the OEM for their first system, is highly uncertain for several reasons. First, the manufacturing is not yet able to take advantage of scale. Second, operational expenses (OpEx) are largely unknown as airborne systems haven't been deployed for sufficiently long trials to fully understand deployment costs and maintenance needs. This LCOE target is approximately 50% larger than that reported by Joshi et al. (2025) for similar size systems in the early commercialization phase, which is consistent with the goal of this work to consider the higher costs of a near-term pilot project.

Here a relatively high OpEx of ~$133 per kW is assumed. This is three to four times the value expected for a towered onshore wind farm. This helps account for operational and maintenance uncertainties. The fixed charge rate (FCR) is assumed to be 8%, which is higher than typically used for wind energy projects and reflects that AWES are not yet mature technologies. The LCOE is given by

$$LCOE = \frac{(CapEx \times FCR) + OpEx}{AEP}$$

Assuming the ¢26/kWh rate target from the OEM and the values given above, this can then be inverted for CapEx once the AEP is calculated. The CapEx is used in MDT to optimize the microgrid design, balancing reliability metrics and costs.

### 3.2 Microgrid Modeling

Two classes of microgrid configurations are considered for 20 year operations. The first is a wind-only configuration including between 1 and 4 AWES and an optional BESS. A second configuration considers the observed complementarity between the solar and wind resources and optimizes a hybrid system including between 1 and 2 AWES, a solar array and an optional BESS.

### 3.2.1 Operation Envelope

Additional factors for the MDT model include the rated cut-in and cut-out wind speeds of the AWES. If the wind speed is too low, the system will be unable to launch or operate; conversely, if the wind speed is too high, there is a risk of damage to the system. According to data sheets published by SkySails, the Venyo system has a cut-out wind speed rating of 25 m/s and a minimum launch/operational wind speed of 5 m/s. Therefore, it is crucial to identify occurrences of wind events outside the operating envelope and evaluate their average frequency throughout the year.

Although, the operational altitude of the system was set to 300-meters for this case study, for cut-in assessment the wind speed at 100-meters is considered for a more conservative approach. Results across seven years (2014-2020) of hourly wind speed data show the year 2020 had the most low-wind speed events, highlighted in Figure 15. That year had over 1,000 hours of low wind speed conditions below the cut-in of 5 m/s. Similarly, high-wind speed events are also assessed across the seven years of data. Results show a single year experienced wind events greater than 25 m/s, with both events occurring in September 2017




as shown in Figure 16. This aligns with the passages of hurricanes Irma and Maria that year (NOAA, 2025). These frequencies of the worst case scenario cut-in and cut-out wind speed events are then input to MDT.

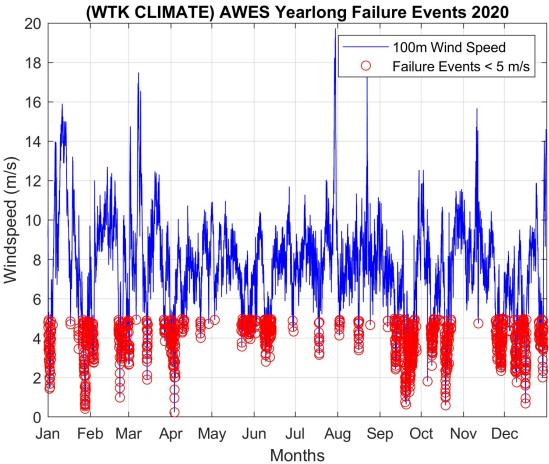

**Figure 15.** Culebra 2020 low-wind speed events below 5 m/s marked in red based on WTK-LED-Climate wind speeds at 100-meter elevation (Draxl et al., 2024).

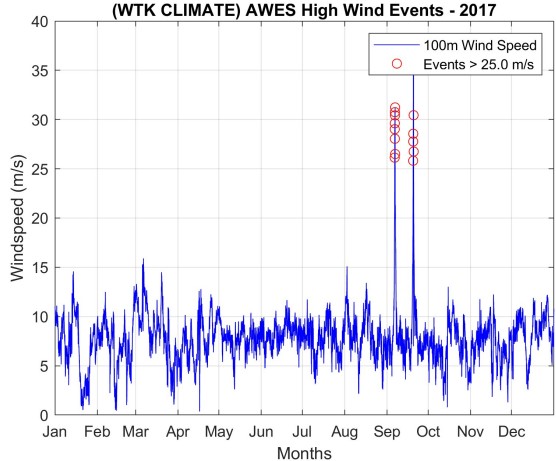

**Figure 16.** Culebra 2017 high-wind speed events above 25 m/s cut-out marked in red using WTK-LED-Climate wind speeds at 100-meter elevation (Draxl et al., 2024).





### 3.2.2 Loads and Target Metrics

Load profiles shown in Figure 12 are added to MDT to simulate a representative rural community for Culebra. This includes the high-priority load of a small health clinic. Residential loads are grouped into blocks 1 through 5 to represent the various groups of homes forming the small rural community.

The performance metric focuses on optimization for the most resilient outcome. Residential loads are set with the non-priority load metric, requiring the microgrid to supply at least 75% of the electricity demand, and targeting a goal of 80%. Similarly, the critical load was set with the priority load metric. This metric requires the microgrid supply a minimum of 90% of the demand, with a goal of 99%. This load is considered critical for the community healthcare needs.

### 3.2.3 Threat Considerations

For the design basis threat modeling, outage frequencies and durations are required for each corresponding threat along with their average time of year occurrence. Using these parameters, representative power outages are simulated in MDT for each corresponding threat event. According to NOAA, during the 15-year interval 2005-2020 there were a a total of 10 tropical storms/depressions, 4 minor hurricanes (category 1-2), and 3 major hurricanes (category 3-5) (NOAA, 2025). Assuming linear distributions for each event, tropical storms were set to occur once per year, minor hurricanes once every 3 years, and major hurricanes once every 5 years. For the average time of year of occurrence, data from NOAA was utilized to set the event interval ranges. The average event timing for tropical storms was centered near the end of August, with the possibility of said event occurring between late July and late September. The average event interval for minor hurricanes was set around mid-to-late August with a range of occurrence from late July to early/mid-September. Finally, for major hurricanes the event interval was centered around early August with a range of occurrence from late July to mid/late-August.

Having set the frequency and interval, outage durations are now required. Unfortunately, PREB excludes major event days from the published grid reliability metrics. Additionally it is important to consider that grid restoration is not a uniform process across feeders or municipalities. Therefore, due to this data gap, the following stress-test assumptions are set for a remote case: a two-week outage for tropical storms/depressions, 30 days for a minor hurricane, and 75 days for a major hurricane. These outage durations are consistent with documented multi-week outages such as the case for tropical storm Ernesto in 2024 where tens of thousand of customer were affected a week after its passage (Coto, 2024), making a two week outage for a rural scenario reasonable. The same can be said for minor hurricanes based on hurricane Fiona in 2022 which disabled the entire grid for as long as 4 weeks in parts of the island (DOE, 2025). Finally, after major hurricanes such as Irma and Maria in 2017, it took 11 months to restore power to all customers around the island, making a 75 day outage a reasonable scenario (GAO, 2019).

For other design basis threats, utility side outages are also included by using published grid reliability metrics from the 2023 Metrics Summary Report to create a representative profile (Puerto Rico Energy Bureau (PREB), 2023). Following the SAIFI trends shown in Figure 3, on a system level, approximately seven power outages occur per month on average. Considering that not all system-level power outages will affect the same municipalities or customers simultaneously, the outage frequency is simplified to one event a week for modeling purposes. From the CAIDI trend from Figure 4, the average power outage duration





**Table 1.** Microgrid configuration (number of AWES and size of BESS) and performance results considering optimistic and conservative AWES losses.

| Assumed AWES Losses (%) | AWES Units (#) | BESS Rating (kW) | NPL (%) | PL (%) | TIC (kW) | BESS Capacity (kWh) |
|---|---|---|---|---|---|---|
| 18% | 3 | 50 | 92.9 | 91.6 | 410 | 200 |
| 27% | 3 | 100 | 92.2 | 91.0 | 460 | 400 |

NPL: Non-Priority Load energy availability, PL: Priority Load energy availability, TIC: Total Installed Capacity.

experienced by customers ranges from over 3.33 hours to under 3 hours, therefore a fixed representative duration of 3.13 hours is selected.

## 4 Results and Discussions

Configurations resulting from the MDT optimization include several with performance just below the compliance metrics and several above, to allow the designer to best interpret the optimum and consider the cost of adding operating margin. First configurations with AWES and a BESS are considered in Section 4.1. Then a solar PV array is additionally incorporated in Section 4.2.

### 4.1 Wind-only Configuration Results

The first simulated cases consider wind-only configurations with an optional BESS. Between 1 and 4 AWES are allowed in the design space. Allowed BESS ratings range from 50-kW to 375-kW discharge and charge rates in 25-kW increments with a fixed 4-hr capacity. Due to the loss uncertainties of AWES technologies, two cases are considered for each configuration, one assuming 18%, and the other 27% total losses. Using these inputs, MDT generated a series of Pareto solutions balancing performance fitness (energy availability) and predicted cost fitness (total capital expenditure).

Results shown in Figure 17 demonstrate a total of eight feasible configurations (green) and eleven infeasible configurations (red), while the 27% loss case produced seven feasible configurations as shown in Figure 18. Each green dot in the Pareto diagram represents a candidate layout for satisfying the proposed metric. However, only the first viable configuration, indicated by the dot highlighted in blue, was assessed. This configuration represents the least-oversized, or most tailored fit for the community load, where other feasible options along the Pareto front increase the total installed capacity, which would result

in a cost-inefficient system. Note that the arrangement of loads shown in Figures 17 and 18 is arbitrarily displayed because no transmission losses or interruptions are modeled - in reality the critical load would be placed nearest the AWES and BESS as shown but those effects are not considered in the modeling.

Table 1 summarizes the performance results for both loss cases. Under the 18% loss assumption, the optimal configuration was composed of three AWES units (120 kW each) and a 50 kW BESS (200 kWh capacity), resulting in a total installed


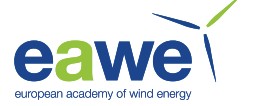
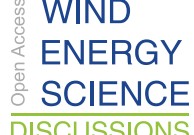

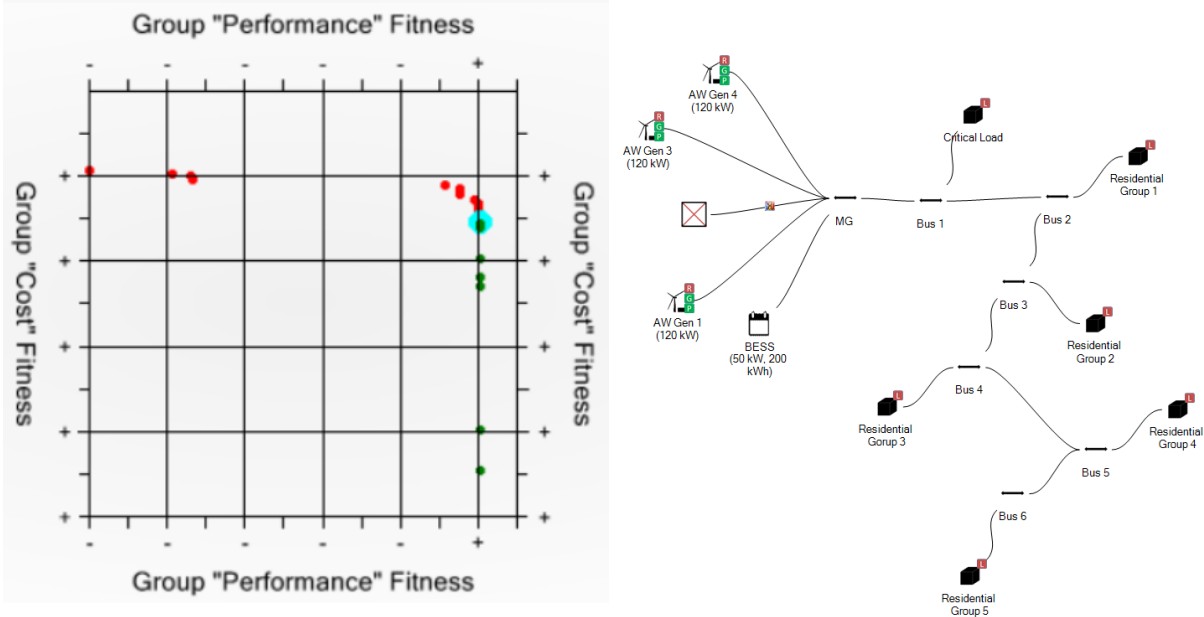

**Figure 17.** Pareto diagram for wind-only configuration results demonstrating eight feasible options (system performance increases to the right, and costs increase going down), and first feasible configuration diagram under 18% AWES losses assumption.

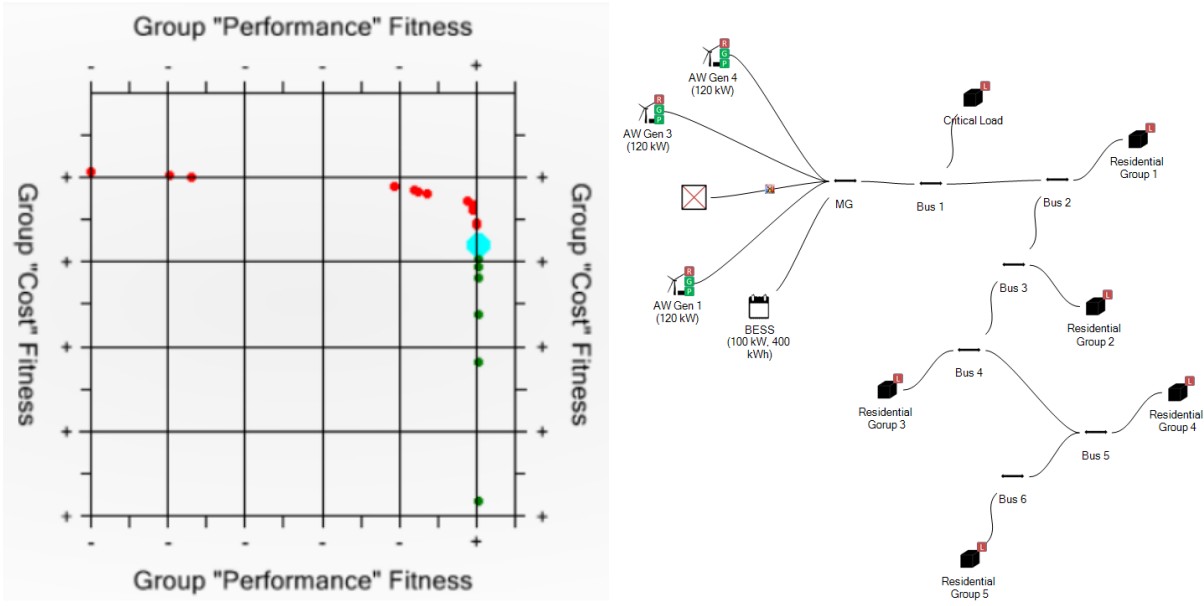

**Figure 18.** Pareto diagram for wind-only configuration results demonstrating seven feasible options (system performance increases to the right, and costs increase going down), and first feasible configuration diagram under 27% AWES losses assumption.





**Table 2.** Microgrid configuration (number of AWES, size of BESS, size of PV array) and performance results considering optimistic and conservative AWES losses for AWES–PV–BESS hybrid configurations.

| Assumed AWES Losses | AWES Units | BESS Rating | PV Rating | NPL | PL | TIC | BESS Capacity |
|---|---|---|---|---|---|---|---|
| (%) | (#) | (kW) | (kW) | (%) | (%) | (kW) | (kWh) |
| 18% | 1 | 50 | 100 | 84.4 | 90.8 | 270 | 200 |
| 27% | 1 | 75 | 125 | 85.6 | 91.1 | 320 | 300 |

NPL: Non-Priority Load energy availability, PL: Priority Load energy availability, TIC: Total Installed Capacity.

capacity of approximately 410 kW. This setup achieved 92.9% and 91.6% energy availability for non-priority and priority loads respectively, demonstrating sufficient coverage to satisfy both previously stated load metric goals of 75% and 90%. The CapEx for the AWES systems in this scenario were ∼ \$495,000 per system to attain ¢26/kWh LCOE.

When the loss was increased to 27% for the second case, MDT compensated by enlarging the BESS to 100 kW (400 kWh capacity) while maintaining three AWES units, increasing the total installed capacity to approximately 460 kW. The resulting energy availability for this case were 92.2% and 91% for both non-priority and priority loads respectively. This preserved energy availability targets despite the reduction in AWES system performance at the cost of a greater BESS rating and higher cost fitness. Because the AEP is reduced by the increased losses, the CapEx for these AWES was only allowed to be ∼\$441,000 per system to attain the same LCOE.

## 4.2 Hybrid Configuration Results

Given the observed wind and solar complementarity in Figure 8, hybrid configurations combining AWES and a solar PV system with an optional BESS were also evaluated. The scenarios of conservative and optimistic losses associated with the AWES were held the same and the same BESS sizing strategy was employed. The PV system sizes considered range from 50kW to 375kW in 25kW increments from one system size to another. Under the 18% loss assumption, MDT produced 16 hybrid configurations (green + blue) as shown in Figure 19. However, only the first configuration represented by the blue dot was selected. Similarly to the wind-only cases, this configuration represents the least oversized system that satisfies the community load requirements established. Table 2 summarizes the the performance metrics for both hybrid cases. For the 18% loss assumption, the first feasible configuration consisted of one 120 kW AWES unit, a 100 kW PV array, and a 50 kW BESS (270 kWh capacity), resulting in a total installed capacity of approximately 270 kW. This system was able to achieve 84.4% and 90.8% energy availability for non-priority and priority loads, respectively meeting both targets of 75% and 90%. For the second case with a loss assumption of 27%, MDT compensated by enlarging both PV and BESS systems to 125 kW PV and 75 kW BESS (320 kWh capacity), while maintaining one AWES unit. This increased the total installed capacity to approximately 320 kW, resulting in slightly higher energy availability levels of 85.6% and 91.1% for non-priority and priority loads respectively.

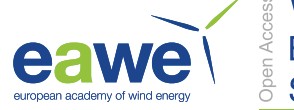



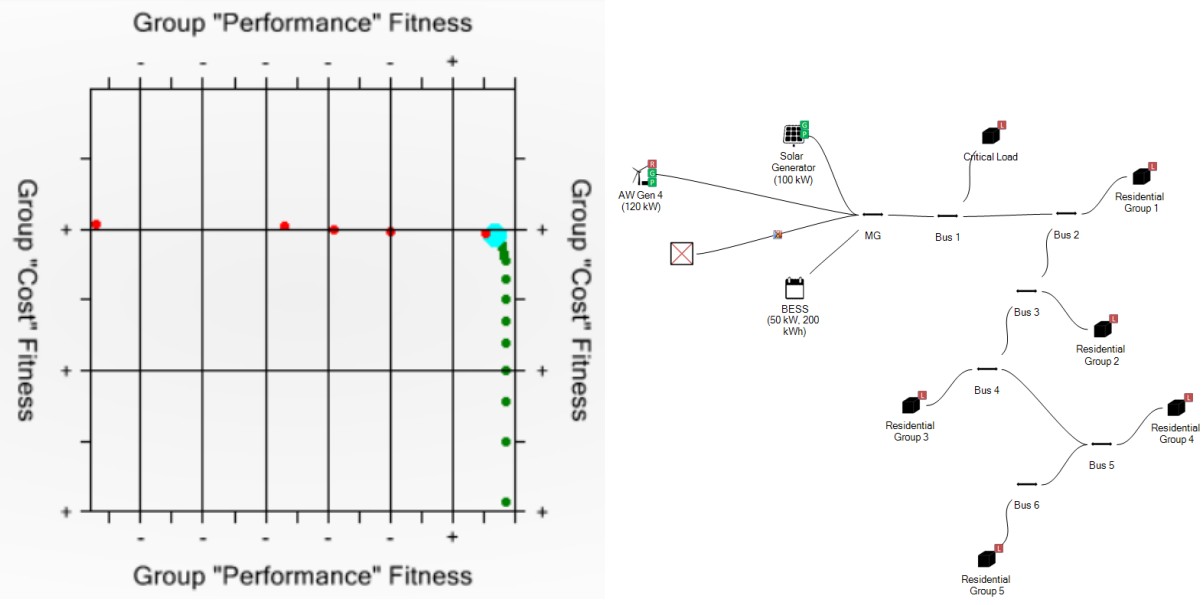

**Figure 19.** Pareto diagram for hybrid configuration results demonstrating 16 feasible options (system performance increases to the right, and costs increase going down), and first feasible configuration diagram under 18% AWES losses assumption.

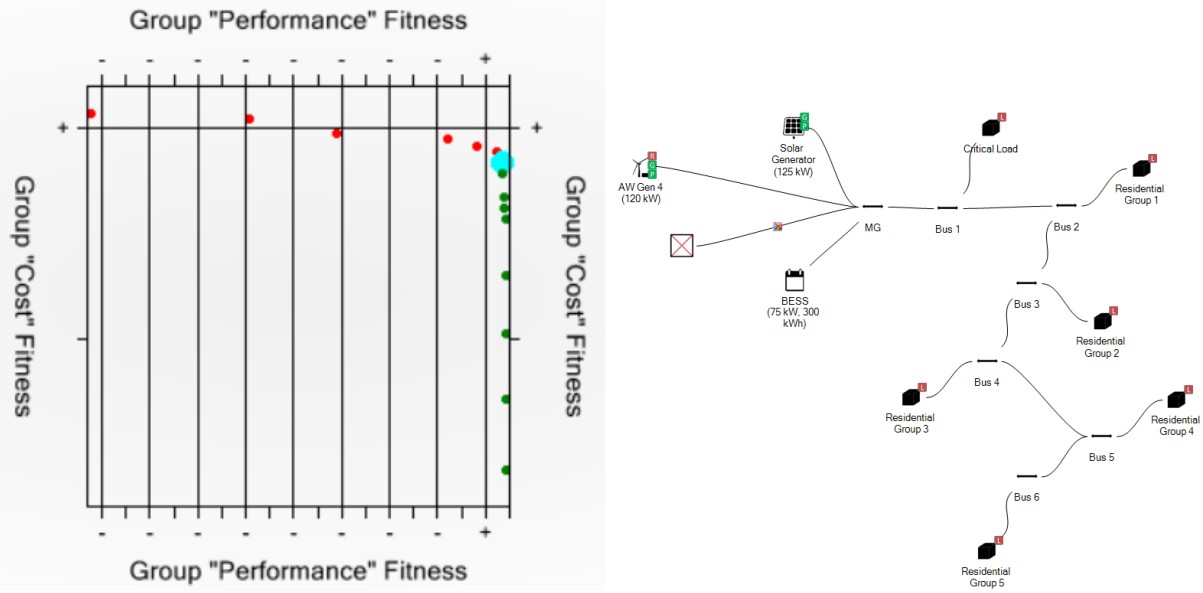

**Figure 20.** Pareto diagram for hybrid configuration results demonstrating 11 feasible options (system performance increases to the right, and costs increase going down), and first feasible configuration diagram under 27% AWES loss assumption





### 4.3 Comparison of Wind-only and Hybrid Microgrids

Comparing the metrics in the tables above, the wind-only system maintains a better non-priority load, but requires three AWES. In comparison, the hybrid system still achieves the target metrics, but requires just one AWES and a modest 100 to 125 kW solar PV array. Both microgrid designs meet the priority load thresholds.

Each microgrid design has advantages and disadvantages. Due to the relatively high cost of the pilot AWES, the hybrid design with just one AWES is much lower cost. This would also provide the opportunity to study the integration of two highly

unsteady energy generation platforms. However, the solar panels in the hybrid design are susceptible to hurricane damage, and thus the hybrid system may be overall less resilient to extreme storms.

In comparison, the wind-only design is theoretically more resilient to storms as the kites would be landed before a storm and the ground stations are robust containers. A significant advantage to the AWE community would come from studying and optimizing an AWE array with the wind-only design. This is a necessary step on the path to future farms of AWES.

## 5 Conclusions

This study evaluated the feasibility and resilience potential of AWES for microgrid applications in Culebra, Puerto Rico using modeled outage events representative of the island's grid conditions, and representative demand data for a small community and clinic. Results show that both configurations are capable of sustaining critical community loads during the modeled outages, however, each configuration has advantages and disadvantages. These systems would also supplement the grid at other times,

at energy rate targets of similar order of magnitude to those currently experienced in Culebra.

The optimized wind-only configuration composed of three AWES units and a BESS achieved energy availabilities of approximately 92-93% and 91-92% for non-priority and priority loads, respectively, with total installed capacities of 410 kW and 460 kW over the range of efficiency loss scenarios assumed for the AWES (18-27%). The 50% increase in assumed losses in the three AWES are compensated by doubling the BESS.

The hybrid configuration integrated one AWES, a solar PV array, and a BESS and achieved energy availabilities of approximately 84-86% and 90-91% for non-priority and priority loads, respectively, with considerably lower installed capacities of 270 kW and 320 kW for 18% and 27% assumed AWES efficiency losses, respectfully. The 50% increase in assumed losses in the one AWES are compensated by just a 50% increase in the BESS.

The wind-only configuration features strong resilience since AWES units can be retracted and stored during hurricanes and

340 severe storms, providing high survivability in comparison to traditional towered wind turbines and fixed PV systems. If kites and tethers were damaged, these could be replaced quickly at modest cost in comparison to replacing towered wind turbines or solar arrays. The higher installed capacity of the wind-only configuration results from robust compliance to the established load performance metrics in combination with the fact that, at the proposed elevation of 300-meters, the AWES are often operating below rated capacity at this site. Therefore, this resilience benefit comes at the expense of a higher expected capital cost due to

345 the larger number of AWES units and higher rated storage requirements. This configuration has the added benefit of providing the opportunity to test multiple AWES functioning together.



In contrast, the hybrid configuration benefits from the favorable wind and solar complementarity of Culebra, where solar production compensates for wind drop events during daytime hours and wind contributes during nighttime periods. This results in improved temporal alignment between generation and demand, smaller storage requirements, a reduced number of AWES units, and therefore a lower required installed capacity and overall system cost. But this hybrid design has a limitation of reduced survivability during extreme weather, as PV arrays remain exposed to hurricane winds and flying debris, which can potentially lead to partial or total system failure despite modern wind resistant mounting systems.

Overall, the results indicate that AWES-based microgrids can considerably enhance energy resilience for islanded and disaster-prone communities, with the wind-only configuration favoring maximum resilience and the hybrid configuration offering a more balanced and economically attractive solution. As AWES technology continues to mature, incorporating verified system cost data, expected lifetime, maintenance frequency, efficiency, and failure rates will enable more accurate levelized cost of energy assessments and support optimized deployment strategies. Future work should also extend the analysis to multi-year simulations and pilot-scale demonstrations to further validate performance under real operating conditions.

*Author contributions.* NEGM was responsible for conceptualization, MDT modeling, interpretation and manuscript preparation. JEQ configured the MDT model. VDG was responsible for mapping in QGIS, EGS developed the technoeconomical analysis and assisted with manuscript review. RDZ was responsible for funding and expertise on Puerto Rico. BCH was responsible for conceptualization, advising on AWE technology, manuscript preparation and funding.

*Competing interests.* The authors declare that they have no any competing interests.

*Disclaimer.* Sandia National Laboratories is a multi-mission laboratory managed and operated by National Technology & Engineering Solutions of Sandia, LLC (NTESS), a wholly owned subsidiary of Honeywell International Inc., for the U.S. Department of Energy's National Nuclear Security Administration (DOE/NNSA) under contract DE-NA0003525.

This written work is authored by an employee of NTESS. The employee, not NTESS, owns the right, title and interest in and to the written work and is responsible for its contents. Any subjective views or opinions that might be expressed in the written work do not necessarily represent the views of the U.S. Government. The publisher acknowledges that the U.S. Government retains a non-exclusive, paid-up, irrevocable, world-wide license to publish or reproduce the published form of this written work or allow others to do so, for U.S. Government purposes. The DOE will provide public access to results of federally sponsored research in accordance with the DOE Public Access Plan.



*Acknowledgements.* This work was supported in part by the U.S. Department of Energy Wind Energy Technologies Office. This paper describes objective technical results and analysis. Any subjective views or opinions that might be expressed in the paper do not necessarily

represent the views of the U.S. Department of Energy or the United States Government.




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
