# Peer review of "Incorporation of Airborne Wind Energy Systems to Enhance Resiliency for a Microgrid in Rural Puerto Rico"

_Wind Energy Science, 2025_

## Referee Comment (RC2)

**Referee comments – Incorporation of Airborne Wind Energy Systems to Enhance Resiliency for a Microgrid in Rural Puerto Rico**

This manuscript provides a fresh and unique view on an interesting application for AWES, focusing on energy resilience. The authors clearly walk the reader through the use-case with a well-chosen case-study in rural Puerto Rico that could be easily repeated for any other location in the world with the same input parameters. The analysis and results of the different optimizations are clear and sound. Therefore, I recommend the manuscript for publication.

A few points could be taken into consideration to elaborate on specific parts of the manuscript:

- Is the storm/hurricane frequency expected to increase in the coming years due to global warming? Which will make a stronger case for AWES compared to towered wind turbines. If this is the case, a linear distribution is chosen for the occurrence of the tropical events in section 3.2.3. Would an exponential distribution be a better fit giving the increasing frequency of tropical storms? This would only have a minor short-term impact but could be something to consider for the long-term value of containerized, stormproof energy systems.
- Although it is true that communities with modest power needs are a good fit for AWES, does it make sense from a deployment and serviceable standpoint? Often early market adaptation requires extra operational support which can prove difficult if these communities are in remote areas, increasing operating costs. At the beginning of section 3.1.1, further maturity of the system is considered. It feels like this contradicts the point of early market adaptation mentioned earlier in the manuscript. Additional clarification would help understanding the logic used.
- It isn't clear why the hourly wind speed is averaged over the 7 years of data in section 3.1.1. By averaging the wind speed, year specific outliers on either side operational wind window are dampened, resulting in a misrepresentation of a real-life scenario. Therefore, letting the simulation run for the full 7 years, calculating the AEP for each year, and then averaging that number, will give a more realistic value. The sentence in 199-200, doesn't clarify enough the reasoning behind averaging the wind speed data before calculating the AEP. It is expected that calculating the AEP and the capacity factor this way will have a minor impact on the manuscript's key results, but it is still something to consider.
- Could the AWES costs for the hybrid configurations added? A similar breakdown as in section 4.1 is expected in 4.2 given that in 4.3 a cost comparison is made.

Furthermore, a couple of technical corrections to improve the readability of the manuscript:

5 – AWES is used for airborne wind energy systemS, however, in this line, a single system is referred to. Look for consistent use of this abbreviation throughout the manuscript (like line 12: "AWES systems").

13 – Sentence is very long and therefore difficult to follow. Consider starting a new sentence from "while a hybrid…."

52 – Use the Euro symbol (€) to stay consistent with the rest of the manuscript.

55 – AWES are **not** only kite based. Other methods of energy generation make use of composite aircrafts for example. To the reader, it isn't clear yet why a kite is mentioned in this line. Same is true for line 154. Only in section 3.1, a kite-based system is introduced.

57 – The final argument is, in my opinion, the strongest case for deployment of AWES in these regions. It follows logically from the introduction before, and I would therefore expect it to be the first driver mentioned in this section. Consider mentioning this driver first.

166 – AWES is plural, is → are. Alternatively, airborne wind energy system could be abbreviated as AWES with the plural AWESs (same for line 173)

179 – Is the cutout wind speed for the system used in section 3.1 considering gusts? If so, 20 m/s sounds reasonable if it assumed that the gusts could push the winds up to 25 m/s. Mentioning that it is on the conservative side in lines 179-180 is in this case not necessary.